# Co-Inoculation of Plant-Growth-Promoting Bacteria Modulates Physiological and Biochemical Responses of Perennial Ryegrass to Water Deficit

**DOI:** 10.3390/plants11192543

**Published:** 2022-09-28

**Authors:** Sandra Cortés-Patiño, Christian D. Vargas, Fagua Alvarez-Flórez, German Estrada-Bonilla

**Affiliations:** 1Rothamsted Research, Protection of Crops and the Environment, Harpenden, Hertfordshire AL5 2JQ, UK; 2School of Biosciences, University of Nottingham, Sutton Bonington LE12 5RD, UK; 3Laboratorio de Fisiología y Bioquímica Vegetal, Departamento de Biología, Facultad de Ciencias, Universidad Nacional de Colombia, Carrera 45 #26-85, Bogotá 111321, Colombia; 4Corporación Colombiana de Investigación Agropecuaria-Agrosavia, Centro de Investigación Tibaitatá, Kilómetro 14 vía Mosquera-Bogotá, Mosquera 250047, Colombia

**Keywords:** drought, livestock, forage, endophytic bacteria, photochemistry

## Abstract

Perennial ryegrass is a forage commonly used in temperate regions for livestock feeding; however, its yield is affected by reduced biomass production under water deficit. In a previous study, three co-inoculations of beneficial bacteria were selected based on their ability to promote plant growth under reduced water availability. The aim of this work was to elucidate some mechanisms by which the selected bacteria can help improve the response of perennial ryegrass to water deficit. Ryegrass plants were inoculated with each of the co-inoculations (*Herbaspirillum* sp. AP02–*Herbaspirillum* sp. AP21; *Herbaspirillum* sp. AP02–*Pseudomonas* sp. N7; *Herbaspirillum* sp. AP21–*Azospirillum brasilense* D7) and subjected to water deficit for 10 days. Physiological and biochemical measurements were taken 10 days after stress and shortly after rehydration. The results showed that bacteria had a positive effect on shoot biomass production, dissipation of excess energy, and proline and chlorophyll pigments during the days of water deficit (*p* < 0.05). The leaf water status of the inoculated plants was 12% higher than that of the uninoculated control after rehydration. Two *Herbaspirillum* strains showed greater potential for use as biofertilizers that help ameliorate the effects of water deficit.

## 1. Introduction

Water deficit is one of the major challenges in agriculture. The variability in water shortage events is expected to continue to increase as a result of global warming; simultaneously, water demand for crop production will increase to meet food security goals [1]. Consequently, there is an urgent need to improve plant water-use efficiency.

Perennial ryegrass (*Lolium perenne* L.) is a forage grass widely used for livestock feeding owing to its high yields, palatability, and digestibility [2]. The shallow root system of this grass makes it highly susceptible to water deficit conditions [3], compromising its yield owing to the large amounts of water needed for its optimal growth [4]. As water deficit progresses, physiological and biochemical processes of plants are disrupted, leading to a decrease in the photosynthetic rate, and thus in the production of new leaves [5]. This is of special importance in ryegrass as a reduction in leaf growth decreases the available food for animal feeding. Studies aiming to find traits linked to drought tolerance in this grass have found great variability within and between cultivars. Several studies have found that the accumulation of compatible solutes like sugars and proline is characteristic of more water-deficit tolerant ryegrass cultivars [2,6], along with an increase in length and root biomass, associated with a more efficient use of water [7]. The ability to absorb energy to be used in photochemistry has also been associated with drought tolerance in ryegrass, as a study with 104 genotypes showed that tolerant genotypes maintained a relatively higher potential quantum efficiency of photosystem II (Fv/Fm) under stress compared with the susceptible genotypes [4]. To help ryegrass plants increase their tolerance to water stress, different approaches have been applied, including the use of mutualistic associations with beneficial microorganisms, like the endophytic fungus *Epichloë* [8,9,10] and, more recently, plant-growth-promoting bacteria (PGPB) [11,12].

The PGPB improve plant growth and development. The mechanisms used by these bacteria to stimulate plant growth can be classified as direct—increasing nutrient availability and the synthesis of growth regulators—or indirect—decreasing the effect of pathogens and pests via the production of antibiotics, competition, and induced systemic resistance, among others [13,14,15]. Furthermore, these bacteria have been proven to enhance the response of plants to abiotic stresses like drought and salinity. The main mechanisms that have been reported to improve plant tolerance to adverse environmental conditions include the production of growth regulators [16,17] and exopolysaccharides [18,19], the activity of the enzyme ACC deaminase [17,20], and the production of volatile compounds that induce systemic tolerance in plants [21]. Their use has widely increased as they have shown to be effective in alleviating abiotic stress in a wide variety of crops; nonetheless, the effect of these bacteria in plant physiology and biochemistry under stress is poorly understood, and more research is needed to decipher the mechanisms involved in the modulation of the stress response of plants. This way, the selection of bacteria could be targeted at the specific conditions of the stress. In grasses, the use of PGPB from the *Bacillus* genera has been shown to increase foliar biomass, chlorophylls, and relative water content, as well as regulate cell wall damage [12,22] and antioxidant response [23].

In a previous work, Cortés-Patiño et al. [24] tested the individual and dual ability of four bacterial strains (*Azospirillum brasilense* D7, *Herbaspirillum* sp. AP02 and AP21, and *Pseudomonas* sp. N7) to promote the growth of perennial ryegrass under water deficit. Three co-inoculations (AP02–AP21, AP21–D7, and AP02–N7) increased aboveground biomass production by more than 50%. Hence, the aim of this work was to investigate the effect of these bacteria in some physiological and biochemical responses to water deficit in inoculated plants. Therefore, a greenhouse experiment was conducted with perennial ryegrass inoculated with the three consortia mentioned above. Plants were subjected to 10 days of water deficit and, to test early recovery, irrigated for three days after the stress. We hypothesized that these consortia could modify the plant water status and photosynthetic activity, which could explain the growth promotion observed previously.

## 2. Results

### 2.1. Biomass Production under Water Deficit

Perennial ryegrass plants subjected to water deficit were mostly affected in their root biomass length and weight. As observed in Table 1, non-irrigated plants showed a significant (*p* < 0.05) reduction in root weight regardless of whether or not the plants were co-inoculated with PGPB. In contrast, root length did not seem to be affected by water stress, but it was affected by the presence of bacteria: inoculation with the combined *Herbaspirillum* strains AP02 and AP21 showed a decrease in root length of around 22.5% when compared with the non-irrigated control plants. Aboveground, no significant differences were observed in the dry biomass of irrigated and non-irrigated plants, but inoculation with AP21–D7 seemed to reduce the production of shoot biomass, while AP02–AP21 increased it. Furthermore, this treatment also increased the number of leaves and tillers by 30%. Overall, the multiple comparison test (Appendix A) showed that the plants inoculated with AP02–AP21 were significantly different from the plants in the other treatments. This difference was mainly due to the higher number of leaves and the increment in aboveground biomass production per root biomass, as can be observed in Table 1 and Appendix A. The other two bacteria treatments, AP02–N7 and AP21–D7, showed no significant differences from the non-irrigated control.

### 2.2. Plant Physiological Response under Water Deficit and after Rehydration

Water stress affected the photochemistry of perennial ryegrass, as observed with the Photosynthesis Yield Analyzer MINI-PAM II, which measures the distribution of light absorbed by the leaves in three different pathways: photochemistry (i.e., charge separation), dissipation as heat, and re-emission as fluorescence. A gradual decrease in the light captured by the leaves was observed after the irrigation was stopped, with a consequent increase in energy loss in non-photochemical processes (NPQ). Although inoculation with bacteria did not significantly affect the maximum quantum yield (Fv/Fm) (data not shown), the calculated distribution of absorbed light (Figure 1) suggests that inoculation with the beneficial bacteria modulated energy loss; more energy was lost by ∆pH- and zeaxanthin-dependent mechanisms, Y(NPQ), than by non-regulated mechanisms, Y(NO), with the AP02–AP21 treatment. On the 10th day of the water deficit, the plants inoculated with this consortium showed 56% energy loss by regulated mechanisms and 17% loss by non-regulated mechanisms, while the non-irrigated control plants showed 46% and 26%, respectively. The other inoculated treatments showed a similar trend, but there was no significant difference with the uninoculated control plants. After rehydration, the quantum yield efficiency YII (i.e., the energy used in photochemistry) did not increase significantly.

Stomatal conductance of plants under water deficit was almost seven times lower than the irrigated control plants, but it was not affected by the inoculation of bacteria (Figure 2A). After rehydration, plants from two PGPB treatments (AP02–AP21 and AP21–D7) showed higher stomatal conductance than the unirrigated control. The loss of water (RWC) in the leaves showed a similar trend—leaves lost between 48.01% and 49.06% in all treatments without irrigation on day 10. Nevertheless, the plants inoculated with PGPB showed a significantly higher RWC compared with the non-inoculated plants after rehydration (*p* < 0.05) (Figure 2B).

### 2.3. Plant Biochemical Response under Water Deficit and after Rehydration

Water deficit was shown to increase the proline and photosynthetic pigment content in the leaves of perennial ryegrass. With respect to proline, the first five days of water deficit showed an almost undetectable increase in content for all treatments (data not shown). On day 10, the content increased significantly (*p* < 0.05) in the plant tissue of all treatments without irrigation (Figure 3A). In two of the inoculated treatments, AP02–AP21 and AP02–N7, the increase was 1.5 times more than in the control without irrigation. In contrast, the proline content in plants from the AP21–D7 treatment was significantly lower (0.65 times). After rehydration, its content decreased drastically in all treatments, becoming almost undetectable. In the irrigated treatment, values close to zero for this compatible solute were observed during the days evaluated, which suggests that the increase in production was directly associated with the water deficit.

Water-soluble sugars (WSSs) showed a constant increase over time regardless of the water deficit treatment. After rehydration, the content decreased for the non-irrigated control and the AP02–N7 and AP21–D7 treatments, but the AP02–AP21 treatment showed a similar concentration to that observed when the plants were under stress (Figure 3B). On day 10, the content of chlorophylls and carotenoids increased significantly for all water deficit treatments (Figure 3C,D). Plants from the AP02–AP21 and AP02–N7 treatments showed the highest pigment content in fresh biomass, with total carotenoid increases of more than 25% with respect to the control without irrigation. In the AP21–D7 treatment, the increase in pigments was less pronounced. After rehydration, a sharp decrease in chlorophyll and carotenoids was observed, but the AP02–AP21 treatment still showed a statistically higher concentration of these pigments (*p* < 0.05).

### 2.4. Correlation between the Physiological and Biochemical Responses of Perennial Ryegrass to Stress

The Pearson correlation analysis (Figure 4) showed a strong negative correlation of stomatal conductance (gs) and RWC with the content of proline (r = −0.8), chlorophyll (r = −0.6), and carotenoids (r = −0.6) in the leaves of perennial ryegrass during the experiment. The dissipation of energy via ∆pH- and zeaxanthin-dependent mechanisms (YNPQ) showed a positive correlation with proline (r = 0.5) and photosynthetic pigment content (r = 0.4) and a negative correlation with gs and RWC (r = −0.4). There was no strong correlation between the energy used in photochemistry, YII, and the biochemical variables evaluated.

## 3. Discussion

In this work, the response of perennial ryegrass to water deficit was characterized by an increase in the concentration of proline, photosynthetic pigments, and NPQ, as well as a reduction in photosystem II activity, stomatal conductance, and RWC. In addition, a decrease in root biomass production was observed.

The effect of water deficit was more prominent in the stomatal conductance of the plants. A reduction in CO_2_ uptake due to stomatal closure can cause damage to the photosynthetic apparatus, as the stomatal response is faster and stomata can close while plants maintain photochemical activity during the stress [25]. In our study, plants subjected to the stress showed a higher dissipation of the energy absorbed by the leaves. The increase in energy dissipated as heat, NPQ, is a photoprotective action that has been correlated with a better tolerance to water deficit in this grass [26]. Notably, plants inoculated with the two *Herbaspirillum* strains (AP02–AP21) showed a significant increase in NPQ, which could indicate a better response of plants to water stress. A greater production of photosynthetic pigments could be associated with this photoprotection effect, as plants inoculated with this treatment also showed the highest concentration of chlorophylls and carotenoids. The photoprotective nature of carotenoids dissipates excess energy in the form of heat to prevent damage to the reaction centers and the formation of reactive oxygen species [27]. The Pearson analysis showed a moderate (r = 0.4) correlation between total carotenoids and non-photochemical quenching (Figure 4). The higher content of pigments in plants inoculated with the bacterial strains could be associated with their ability to fix nitrogen, as these bacteria have been shown to grow in nitrogen-free medium [24] and to improve the nitrogen content in perennial ryegrass [28]. The potential of *Herbaspirillum* to increase the nitrogen content has also been previously reported in crops like maize [29,30].

Although, in our study, an increase in non-photochemical quenching was observed when plants were inoculated with PGPB, other studies have found contrasting results that were also associated with a beneficial effect of the bacteria [31]. Because of the great variability in the responses of plants to water deficit, both an increase and a decrease in heat dissipation can indicate a positive effect of PGPB inoculation. A similar scenario occurs with proline, a compatible solute that accumulates in plants under stress conditions. This solute has been shown to act as an osmoprotective agent for subcellular structures and macromolecules, a molecular chaperone for proteins, and an antioxidant defense molecule [32,33]. Studies aiming to find ryegrass genotypes and phenotypes resistant to drought have frequently found a relationship between the accumulation of proline and resistance to stress [2,4,34]. Nevertheless, the role of proline is a subject of wide debate, and its buildup has been observed in genotypes hypersensitive and tolerant to osmotic stress [32]. In this work, the production of proline differed according to the PGPB treatment, but, as observed with photosynthetic pigments, the co-inoculation of AP02–AP21 and AP02-N7 showed the highest content of this solute, and AP21-D7 showed the lowest. The ability of PGPB to modulate proline content in plants under stress, both increasing and decreasing its concentration in plant tissue, has been interpreted as a mechanism by which bacteria help diminish the severity of stress in tomato, maize, and rice plants, among others [29,35,36].

After the rehydration, proline levels decreased drastically in all treatments subjected to the stress, as this osmolyte catabolism provides energy to resume growth once the optimal conditions are re-established [32]. The content of this solute in the leaves showed a strongly negative correlation with stomatal conductance and RWC (r = −0.8), both of which rapidly recovered when the water supply was restored. According to Yates et al. [5], perennial ryegrass rapidly resumes leaf growth after rewatering. This could be due to the “state of preservation” reported by Wilson [37]. During water deficit, the authors observed arrested growth, meaning that the plants did not reinvest the photo assimilates for the production of biomass, but for the maintenance of key functions to recover its development when the water potential was restored. Notably, all plants inoculated with PGPB showed a significantly higher capacity to recover leaf RWC when water was supplied. The ability of PGPB to improve RWC in plants subjected to water deficit has been mainly related to an increase in growth regulators, such as auxins and abscisic acid, in plants, as well as the production of compatible solutes such as proline [35,38,39]. In our study, the bacteria did not improve RWC during the stress, but after rehydration, which can also be linked to the ability of roots to increase water uptake.

Of the three co-inoculations, the combination of *Herbaspirillum* sp. AP02 and *Herbaspirillum* sp. AP21 allowed for a higher production of foliar biomass in ryegrass plants, while the root biomass was significantly lower than in the other treatments. A decrease in root biomass was also observed by Neiverth et al. [40] in wheat plants inoculated with *Herbaspirillum seropedicae*. The authors observed that the plants had a greater amount of root hairs and a higher production of foliar biomass. It is suggested that this effect of PGPB is mainly due to their ability to produce indolic compounds, as high concentrations of these compounds stimulate the formation of lateral roots, decrease the length of the main root, and increase the formation of root hairs [41]. In this way, the secondary roots and root hairs allow for a greater uptake of nutrients for the production of foliar biomass. This could potentially explain the higher RWC in the leaves of plants after rehydration, but further work with these PGPB is needed to observe whether their inoculation changes the root morphology and architecture in perennial ryegrass. Furthermore, the increase in aboveground biomass of plants inoculated with AP02–AP21 is one of the most reported effects due to the inoculation of PGPB, and has been observed in crops under environmental stress, such as drought, frost, and metal toxicity [14,42]. It is believed that this effect is mainly due to the ability of PGPB to increase the availability and uptake of nutrients and to produce growth regulators [43].

The results obtained in this work suggest that the co-inoculation of the two *Herbaspirillum* strains (AP02–AP21) has a positive effect in perennial ryegrass under water deficit and helps the plants to recover faster. This co-inoculation has shown the ability to increase the dissipation of energy as a photoprotective mechanism. Furthermore, these bacteria showed the ability to promote biomass production aboveground with a low root biomass, suggesting an effect of growth regulators produced by these bacteria. In a previous work [24], we demonstrated the potential of these bacteria to produce indolic compounds and exopolysaccharides, colonize plant tissue, and stimulate root growth through volatile production, and these traits could play a key role in the modulation of the response of ryegrass to water deficit and rehydration. Overall, our results demonstrate the ability of the co-inoculation AP02–AP21 to modify plant physiology and biochemistry to improve the response of perennial ryegrass to water deficit.

## 4. Materials and Methods

### 4.1. Bacterial Strains and Consortia

The bacterial strains used in this study were provided by the collection of microorganisms of the Colombian Corporation for Agricultural Research (Agrosavia). Three bacterial consortia were tested in this experiment based on previous work by Cortés-Patiño et al. [24]: (i) *Herbaspirillum* sp. AP02 and *Herbaspirillum* sp. AP21, (ii) *Herbaspirillum* sp. AP02 and *Pseudomonas fluorescens* N7, and (iii) *Herbaspirillum* sp. AP21 and *Azospirillum brasilense* D7. Each strain was separately multiplicated by inoculating one bacterial colony in 250 mL flasks containing DYGS liquid medium [44] (g L^−1^: glucose, 2.0; malic acid 2.0; peptone,1.5; yeast extract, 2.0; K_2_HPO_4_, 0.5; MgSO_4_.7H_2_O, 0.5; and glutamic acid, 1.5) and incubating them at 150 rpm and 30 °C for 48 h. A bacterial population of 10^8^ colony forming units mL^−1^ was obtained for each strain and the bacterial broths were mixed immediately before plant inoculation in equal volumes (1:1 v v^−1^).

### 4.2. Greenhouse Experiment

The response of perennial ryegrass to water deficit was evaluated under semi-controlled conditions in a greenhouse facility at the Tibaitatá Research Center of Agrosavia in Mosquera, Cundinamarca, Colombia (4°41′43.6″ N latitude, 74°12′19.9″ W longitude; 2600 m above sea level). The environmental conditions were monitored with a datalogger (U23-001, HOBO, Bourne, USA) during the course of the experiment. The temperature and relative humidity of the greenhouse ranged from 12.5 °C to 24.6 °C and from 46% to 83%, respectively.

Seeds of perennial ryegrass were surface-disinfected with 3% sodium hypochlorite (v v^−1^) (2 min) and 70% ethanol (v v^−1^) (3 min) and then washed repeatedly with sterile distilled water. The seeds were then placed in nutritive gelified medium [45] and incubated in the dark at 28 °C for five days for germination. Following this, three germinated seeds were planted in pots containing 2 kg of unsterile soil. The soil is classified as Vitric Haplustands-AMBc of the order andisol (Soil Science Division Staff, 2017) [46]. The soil properties were as follows: pH 6.04, organic matter (130 g kg^−1^), effective coefficient for cationic exchange (28.39 cmol kg^−1^), P (381 mg kg^−1^), K (4.18 cmol kg^−1^), S (23.90 mg kg^−1^), Ca (15.95 cmol kg^−1^), Mg (8.78 cmol kg^−1^), Na (0.40 cmol kg^−1^), and Fe (620.03 mg kg^−1^).

After seven days of growth, two seedlings were removed and only one was maintained in each pot. Subsequently, the pots were inoculated with 5 mL of each bacterial consortia (except for the irrigated and drought control treatments) and the pots were maintained close to the field capacity (90%). Leaves were cut to 6 cm after 21 days of growth to simulate grazing and then left to grow at normal irrigation for 14 days before the start of the drought treatment. Irrigation was completely stopped for 10 days and then re-established for 3 days to test early recovery. Physiological and biochemical parameters were measured for both stress and recovery. At the end of the recovery sampling, the number of tillers and leaves, shoot and root length, and dry biomass of the plants were quantified.

Each inoculated treatment consisted of 11 pots. Five pots were selected for measurements of chlorophyll fluorescence and stomatal conductance for both water stress and recovery. For the RWC measures and biochemical analysis, a destructive sampling of three pots per treatment was performed each time. The experiment was performed twice with similar responses, so the analyses integrated the data from the two independent experiments.

### 4.3. Determination of Physiological Parameters

Five plants from each treatment were selected at the start of the experiment for chlorophyll fluorescence and stomatal conductance measurements. From these plants, fully developed and healthy leaves were selected. Chlorophyll fluorescence measures were taken with a photosynthesis yield analyzer (MINI-PAM II, Walz, Effeltrich, Germany) in dark-adapted leaves from 8 am to 10 am each day. Stomatal conductance was measured with a leaf porometer (SC-1, Meter Environment, Pullman, USA) every two hours from 6 am to 4 pm.

The RWC was estimated according to the methodology of Sade et al. [47] for leaves without petiole. In brief, segments of 6 cm of leaves were taken from each plant and placed inside zipper-locked plastic bags—previously weighted—to determine their fresh weight. Subsequently, 2 mL of 5 mM CaCl_2_ was added to each bag and the bags were left in the dark for 8 h. Then, the leaf segments were weighed (turgid weight) and placed in an oven at 60 °C for 72 h (dry weight). These values were used to calculate the relative water content of each leaf using the following equation:$$RWC\;\left(\%\right)=\frac{\left(Fresh\;weight-Dry\;weight\right)}{\left(Turgid\;weight-Dry\;weight\right)}\times 100$$

### 4.4. Determination of Biochemical Parameters

Proline and WSS were measured throughout the water deficit period and after rehydration. Proline concentration was quantified using the methodology described by Bates et al. [48], with the modifications of Moreno-Galván et al. [23]. Briefly, 100 mg of fresh leaf tissue was homogenized in 1 mL of 3% sulfosalicylic acid (w v^−1^) and centrifuged at 10,000× *g* for 5 min (Eppendorf 5415C Centrifuge, Hamburg, Germany). Then, 100 µL of the supernatant was mixed with 100 µL of sulfosalicylic acid (3% w v^−1^), 200 µL glacial acetic acid, and 200 µL acid ninhydrin solution (1.25 g ninhidrin, 30 mL of acetic acid 20 mL, and 6 M phosphoric acid). The mix was incubated at 100 °C for one hour and the reaction was stopped in an ice bath before adding 1 mL of toluene. After homogenizing with vortex, 200 µL of the aqueous phase was taken for absorbance reading at 520 nm in a microplate reader (Synergy HTX, Multi-Mode Reader, BioTek Instruments, Inc., VT, Santa Clara, CA, USA). Proline concentration was compared against a standard curve with L-proline and calculated based on fresh weight (mg g^−1^ fresh biomass).

The WSS concentration was quantified from fresh leaf tissue. The extraction was performed from 100 mg of ground leaf tissue by adding 5 mL of distilled water and incubated at room temperature in a horizontal shaker for 1 h. The mix was then centrifuged at 10,000× *g* for 7 min at 12 °C, and the supernatants were collected in clean tubes for the quantification of water-soluble carbohydrates using the phenol-sulfuric acid methodology by Dubois et al. [49]. In brief, 30 µL of the supernatant was mixed with 180 µL of distilled water, 200 µL of 80% phenol (w v^−1^), and 1 mL of 98% sulfuric acid (w v^−1^). The mix was then vortexed and cooled at room temperature. Absorbance was determined at 490 nm in a microplate reader. The concentration of water-soluble carbohydrates was determined on a standard curve with D-glucose and calculated based on fresh weight (mg g^−1^ fresh biomass).

Chlorophyll and carotenoid contents were measured according to the methodology described by Rojas-Tapias et al. [50]. Briefly, 10 mg of fresh tissue was mixed with 1 mL of dimethyl sulfoxide in a microtube and incubated at 95 °C for 2 h. Then, absorbance was read at 665.1 nm, 649.1 nm, and 480 nm in a microplate reader, and the readings were replaced in the equations described by Wellburn [51]. The concentrations of chlorophyll a and b and carotenoids were expressed in concentrations based on fresh weight (mg g^−1^ fresh biomass).

### 4.5. Determination of Plant Development Parameters

The number of tillers and leaves in each plant was determined following rehydration. The dry weight was determined after drying the root and shoot biomass in a dry heat incubator for 72 h. After this, the root-to-shoot ratio was determined based on the dry weight. Root and shoot lengths were also determined.

### 4.6. Statistical Analysis

Statistical analysis was carried out using R Studio software (v3.6.1). The plant morphological parameters were analyzed via gDGC multiple comparison of means with significance level α = 0.05 and ward linkage and validated with a Hotelling test with Bonferroni correction (*p* > 0.05). The heatmap was created using the pheatmap package [52]. Conductance, physiological, and biochemical parameters were assessed using a generalized linear mixed model (GLMM) with Gamma distribution and log-link function—the replicates were used as the random part, the fixed part of the model corresponded to treatments, and multivariate validation was performed by ANOVA simultaneous component analysis (ASCA) [53]. GLMM was generated using the glmm package [54] and ASCA was generated using the MetStaT package [55]. The graphs were constructed using the ggplot2 package (v.3.3.5) and the ggcorrplot package (v.0.1.3) was specifically used for the Pearson correlation matrix.

## 5. Conclusions

In this work, PGPB co-inoculations showed the ability to modulate the severity of the stress response of perennial ryegrass to water deficit. Using a multivariate analysis, we were able to show that the effect of PGPB can have an impact on different morphological, physiological, and biochemical parameters in plants. Plants inoculated with the two *Herbaspirillum* strains showed higher shoot biomass production, increased protection of photosystem II via heat dissipation, and higher proline and carotenoid production. We suggest that these bacteria be used in future works, where root morphology and architecture and net photosynthesis can also be evaluated to help elucidate the mechanisms by which these bacteria help increase the tolerance of ryegrass to water deficit.

## Figures and Tables

**Figure 1 plants-11-02543-f001:**
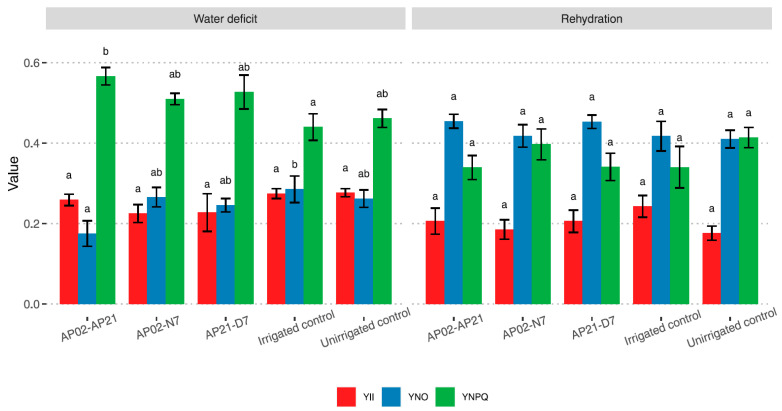
Calculated distribution of the absorbed light in the leaves of perennial ryegrass after 10 days without irrigation and 3 days after rehydration. YII: energy used in photochemistry, YNPQ: energy lost by ∆pH- and zeaxanthin-dependent mechanisms, YNO: energy loss through non-regulated mechanisms. AP02–AP21: *Herbaspirillum* sp. AP02 and *Herbaspirillum* sp. AP21; AP02–N7: *Herbaspirillum* sp. AP02 and *Pseudomonas fluorescens* N7; AP21–D7: *Herbaspirillum* sp. AP21 and *Azospirillum brasilense* D7. Each value represents the mean of the replicates from the two independent experiments ± SE. Different letters indicate statistically significant differences observed according to the Fisher’s LSD test with Bonferroni correction (*p* < 0.05).

**Figure 2 plants-11-02543-f002:**
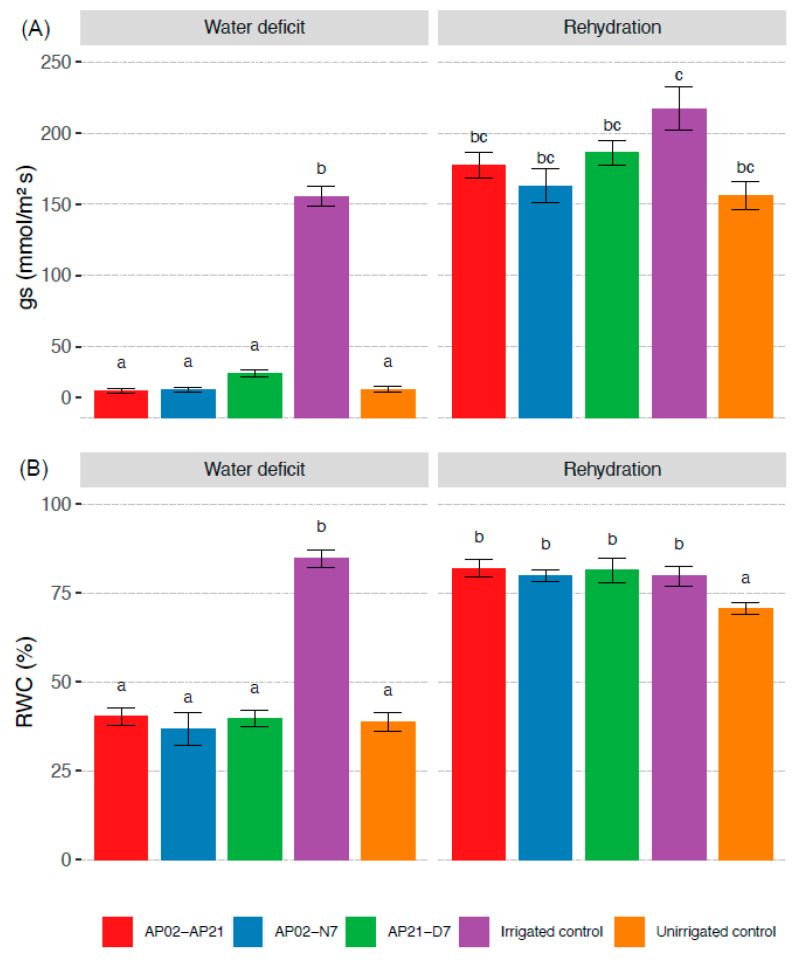
Stomatal conductance (gs) (**A**) and leaf relative water content (RWC) (**B**) in perennial ryegrass after 10 days without irrigation and 3 days after rehydration. AP02–AP21: *Herbaspirillum* sp. AP02 and *Herbaspirillum* sp. AP21; AP02–N7: *Herbaspirillum* sp. AP02 and *Pseudomonas fluorescens* N7; AP21–D7: *Herbaspirillum* sp. AP21 and *Azospirillum brasilense* D7. Each value represents the mean of the replicates from the two independent experiments ±SE. Different letters indicate statistically significant differences observed according to the Fisher’s LSD test with Bonferroni correction (*p* < 0.05).

**Figure 3 plants-11-02543-f003:**
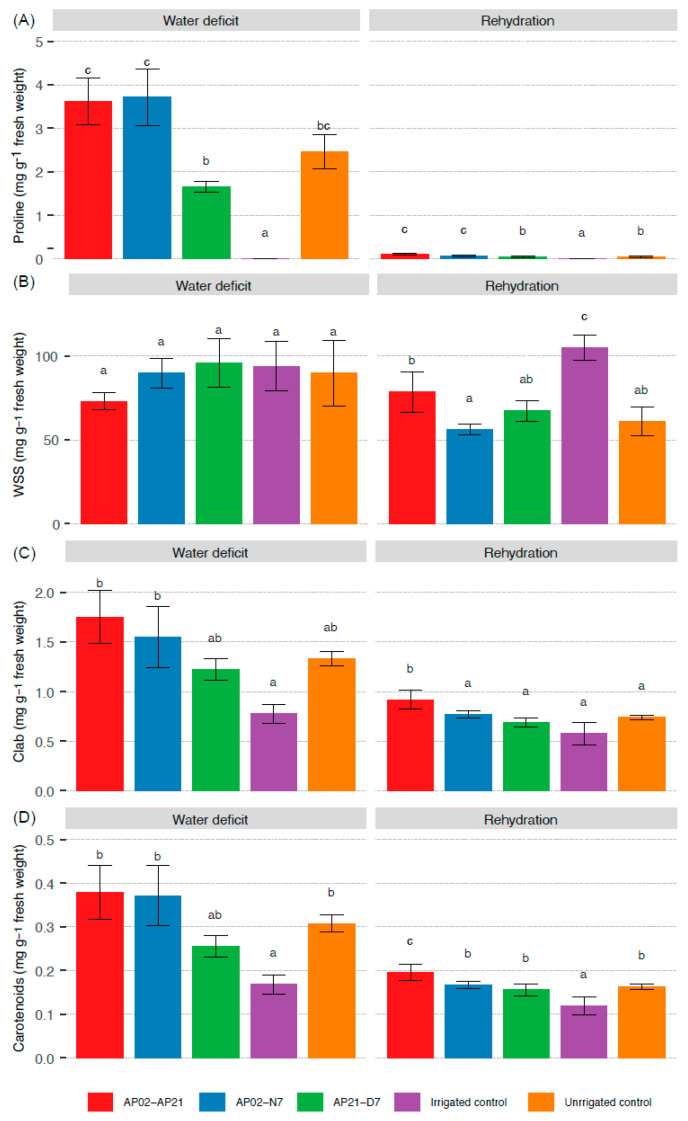
Biochemical parameters of perennial ryegrass after 10 days without irrigation and 3 days after rehydration on proline content (**A**), water-soluble sugars (WSS) (**B**), chlorophyll (Clab) a and b (**C**), and carotenoids (**D**). Each value represents the mean of the replicates from the two independent experiments ± SE. Different letters indicate statistically significant differences observed according to Fisher’s LSD test with Bonferroni correction (*p* < 0.05).

**Figure 4 plants-11-02543-f004:**
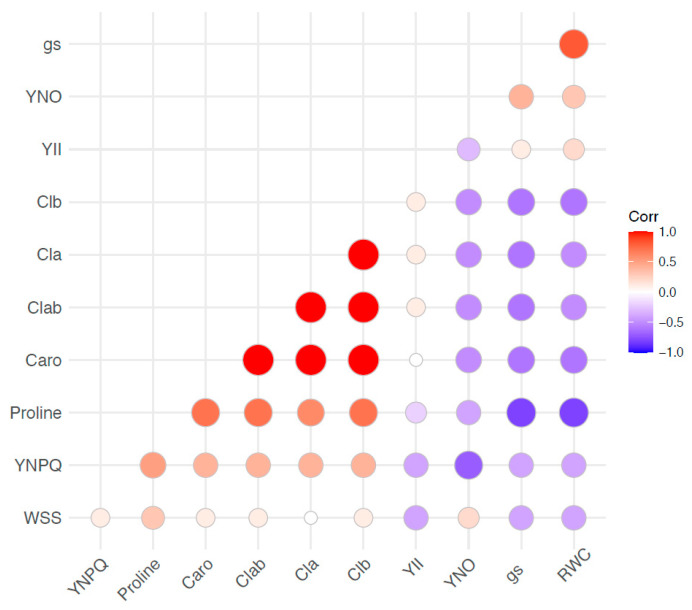
Pearson’s correlation matrix of the physiological and biochemical traits evaluated under water deficit and rehydration. Correlations are displayed based on color intensity (red = positive, blue = negative). Circle size and color intensity are proportional to the correlation coefficient. YNPQ: energy lost by ∆pH- and zeaxanthin-dependent mechanisms, YII: energy used in photochemistry, YNO: energy loss through non-regulated mechanisms, gs: stomatal conductance, RWC: leaf relative water content, WSS: water-soluble sugars, clab: chlorophylls a and b, caro: total carotenoids.

**Table 1 plants-11-02543-t001:** Morphological parameters of perennial ryegrass after rehydration. The plants were subjected to 10 days of water deficit and rehydrated for 3 days. Each value represents the mean of the replicates from the two independent experiments ± SE. Different letters indicate statistically significant differences observed according to the Fisher’s LSD test with Bonferroni correction (*p* < 0.05).

Treatment	Leaf Weight (g plant^−1^)	Root Weight (g plant^−1^)	Leaf Length (cm plant^−1^)	Root Length (cm plant^−1^)	Leaf Number (plant^−1^)	Tiller Number (plant^−1^)	Root/Shoot
AP02–AP21	2.66 ± 0.15 b	1.38 ± 0.04 a	26.80 ± 0.84 a	19.02 ± 0.85 a	106.50 ± 7.24 a	35.00 ± 2.38 a	0.53 ± 0.38 a
AP02–N7	2.32 ± 0.16 ab	1.60 ± 0.06 a	28.44 ± 0.65 a	20.50 ± 1.01 ab	84.80 ± 8.02 a	31.40 ± 4.09 a	0.75 ± 0.02 b
AP21–D7	1.97 ± 0.08 a	1.44 ± 0.04 a	29.52 ± 0.81 a	23.35 ± 0.54 abc	81.83 ± 5.24 a	26.67 ± 2.53 a	0.74 ± 0.04 b
Irrigated control	2.45 ± 0.12 ab	1.98 ± 0.12 b	27.17 ± 2.22 a	26.17 ± 1.65 c	81.83 ± 7.34 a	29.33 ± 3.62 a	0.81 ± 0.04 b
Unirrigated control	2.21 ± 0.06 ab	1.62 ± 0.04 a	29.20 ± 0.57 a	24.55 ± 1.03 bc	81.83 ± 5.64 a	26.83 ± 1.53 a	0.74 ± 0.04 b

## Data Availability

The data presented in this study are available upon request from the corresponding author.

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
