# Peer review of "Co-Inoculation of Plant-Growth-Promoting Bacteria Modulates Physiological and Biochemical Responses of Perennial Ryegrass to Water Deficit"

_plants, 2022, doi:10.3390/plants11192543_

Round 1

Reviewer 1 Report

Revision to

Title: Co-inoculation of plant growth-promoting bacteria modulate physiological and biochemical responses of perennial ryegrass to water deficit

Authors: Sandra Cortés-Patiño, Christian D. Vargas, Fagua Alvarez-Flórez, German Estrada-Bonilla.

Abstract Journal: Plants

Manuscript number: plants-1931858

General remarks: The manuscript by Cortés-Patiño et al. reported an interesting investigation about the ability of different plant growth-promoting bacteria (PGPB) co-inoculation to improve the response to ryegrass to  drought stress. The authors analyzed different abiotic stress marker to evaluate the response to water stress. The manuscript is interesting but requires significant improvements before acceptance. The English language and fluency and the quality of data presentation need substantial corrections to be more clear for readers. Particularly the improvement of English is mandatory for the publication.

 Furthermore, authors should improve discussion sections. The effects of the treatment are well described but the discussion completely lacks of an hypothesis about the mechanism of action of the PGPB in the improvement of the analyzed effects. The results shown by the authors are not sufficient to clarify how PGPBs act on plants drought response but the authors should at least speculate about this. Based on these considerations, the manuscript will be suitable for publication on Plants after major revisions.

Introduction:

The introduction is too much generic in some parts, and should be better developed, e.g. detailing the significance of drought stress for ryegrass cultivation.

Further, also PGPB topic should be better introduced, specifying how these microorganisms report beneficial effects for plant cultivation.

Results:

Authors should better argued their data clarifying the effective significance (certainly of value), improving the comparison of drought treatments stress and differences PGPB co-inoculation mixtures.

Discussion:

See general remarks

Figures and Tables:

A general improvement of figures quality is necessary.

In my opinion Figure 1 should be moved in supplementary while Table 1 should directly show the significant differences.

Author Response

Dear Reviewer.

Reviewer 2 Report

A fun and well crafted paper to read-- thank you!

A few comments  added as sticky notes  but a stronly supported story  

Author Response

Dear Reviewer.

Round 2

Reviewer 1 Report

The R1 version of the manuscript by Cortes-Patino is significantly improved. All the requested comments were revised by the authors. The quality of the manuscript and the English language are now adequate. In my opinion this manuscript is suitable for publication on Plants.